# Cardiovascular Health Peaks and Meteorological Conditions: A Quantile Regression Approach

**DOI:** 10.3390/ijerph182413277

**Published:** 2021-12-16

**Authors:** Yohann Moanahere Chiu, Fateh Chebana, Belkacem Abdous, Diane Bélanger, Pierre Gosselin

**Affiliations:** 1Faculty of Pharmacy, Laval University, 1050 Avenue de la Médecine, Quebec, QC G1V 0A6, Canada; 2TEIAC Unity, Quebec National Institute of Public Health, 945 Avenue Wolfe, Quebec, QC G1V 5B3, Canada; pierre.gosselin@inspq.qc.ca; 3Centre Eau Terre Environnement, Institut National de la Recherche Scientifique, 490 Rue de la Couronne, Quebec, QC G1K 9A9, Canada; Fateh.Chebana@ete.inrs.ca (F.C.); diane_belanger@bell.net (D.B.); 4Department of Social and Preventive Medicine, Faculty of Medicine, Laval University, 1050 Avenue de la Médecine, Quebec, QC G1V 0A6, Canada; belkacem.abdous.1@ulaval.ca; 5Centre de Recherche, Centre Hospitalier Universitaire de Québec, 2875 Boulevard Laurier, Quebec, QC G1V 2M2, Canada

**Keywords:** quantile regression, cardiovascular diseases, health peaks, meteorological conditions, environmental health, heatwaves

## Abstract

Cardiovascular morbidity and mortality are influenced by meteorological conditions, such as temperature or snowfall. Relationships between cardiovascular health and meteorological conditions are usually studied based on specific meteorological events or means. However, those studies bring little to no insight into health peaks and unusual events far from the mean, such as a day with an unusually high number of hospitalizations. Health peaks represent a heavy burden for the public health system; they are, however, usually studied specifically when they occur (e.g., the European 2003 heatwave). Specific analyses are needed, using appropriate statistical tools. Quantile regression can provide such analysis by focusing not only on the conditional median, but on different conditional quantiles of the dependent variable. In particular, high quantiles of a health issue can be treated as health peaks. In this study, quantile regression is used to model the relationships between conditional quantiles of cardiovascular variables and meteorological variables in Montreal (Canada), focusing on health peaks. Results show that meteorological impacts are not constant throughout the conditional quantiles. They are stronger in health peaks compared to quantiles around the median. Results also show that temperature is the main significant variable. This study highlights the fact that classical statistical methods are not appropriate when health peaks are of interest. Quantile regression allows for more precise estimations for health peaks, which could lead to refined public health warnings.

## 1. Introduction

Cardiovascular diseases (CVDs) are the second cause of all deaths behind cancers [1] in Canada. In Québec, they are the second cause for all hospitalizations, whereas cancers are the fifth [2]. The number of hospitalizations due to CVDs has been constantly increasing each year by 2.3% on average since 2005 [3]. Therefore, they represent a major public health challenge. This situation is not uncommon in developed countries, since CVDs are the first cause of death in the world [4]. Therefore, CVDs comprehension, analysis, and prevention remain significant tasks [5,6].

Various factors can influence CVDs. Life habits such as smoking, nutrition or sport practice are known to play important roles [7,8,9]. On the other hand, atmospheric pollution (e.g., ozone or particulate matter) is strongly associated with CVD occurrence [10]. It is now widely accepted that air pollution has a predominant impact on CVDs, both indoor and outdoor [11,12]. In addition, a large body of literature dealing with impacts of meteorological conditions on CVD occurrence such as cold exposure, heatwaves, and diurnal temperature variations is available [13]. Snowfall and humidity are also linked to a higher CVD occurrence [14,15]. Moreover, those impacts are also present in Québec, especially regarding temperature [16,17,18]. However, climate change, by modifying the meteorological variables distribution, may also change those established relationships. In Québec, a significant increase in mean temperatures is, among others, expected by 2050 [19] with the number of hot days (≥30 °C) likely to be multiplied by a factor of three. Consequently, heatwaves, which are linked to CVD occurrences, will increase in intensity and frequency. Thus, it is crucial to properly quantify relationships between peaks and meteorological conditions in order to promote population and health services adaptation.

Most of the available studies use statistical models that implicitly focus on mean health events, not specifically peaks. This is a necessary process in order to understand relationships between a health issue and meteorological variables, notably for CVD surveillance. Nevertheless, health peaks (or simply peaks) are not correctly analyzed by those models. Those specific observations lie far from average events and their modeling using statistical models based on the mean can be misleading (results from mean-based regressions cannot be extended to noncentral locations; [20]). Furthermore, they are by definition scarce and they can have heavy consequences on the health system (e.g., emergency departments overflow [21]), which should warrant studies on their own. Few studies examine the extreme nature of health variables from a statistical modeling point of view and they usually focus on peak series once they have been extracted from the datasets [22,23,24]. Although peaks have strong implications, they are usually not studied from an overall point of view. They can also be analyzed from a descriptive (without modeling) and local point of view for a given extreme event, such as the 2003 European heatwave [25]. Quantile regression (QR) is a statistical model which employs all the observations and would allow comparing the impacts of meteorological variables on average events as well as on health peaks with higher quantile order [26].

The QR model can explore not only the conditional median (as a measure of central tendency), but all quantiles of the dependent variable. Since peaks can be defined as high quantiles, QR is appropriate to investigate in-depth the conditional distribution of peaks. Moreover, an independent variable could have a significant effect in one quantile, whereas it would not be so in another quantile. Studying only the conditional mean actually “dilutes” information related to the rest of the conditional distribution, especially concerning peaks. QR has been used in public health, but not to explore relationships with meteorological conditions (even less so concerning CVDs). QR is popular when it comes to studying indicators that may have heterogeneous effects. In health sciences, QR has shown the ability to discover heterogeneous effects of independent variables such as air pollution or socioeconomic factors on various health issues [27,28,29]. This non-homogeneous influence over different quantiles cannot be highlighted by standard linear regression, which estimates only the conditional mean (as does any other mean-based regression such as the Poisson regression [26]). On the other hand, by using QR, the whole conditional distribution of a health outcome can be reproduced, and comparisons of multiple quantiles can then take place. The objective of this study is thus to examine the impact of meteorological conditions on CVD morbidity and mortality with QR, focusing on health peaks (high quantiles) and their differences with other quantiles, relative to the whole conditional distribution.

The rest of this paper is organized as follows. Available data and methods are described in Section 2, while the results are shown in Section 3. Discussion and limitations are presented in Section 4. Finally, Section 5 presents the conclusions of this study.

## 2. Materials and Methods

This section describes the data and statistical methods used in this study. A more exhaustive description is given in the complete research report of Chiu et al. [30].

### 2.1. Data

The study area is Montreal metropolitan community in Canada (abbreviated as Montreal, Figure 1), with 3,994,990 inhabitants in 2018. In this study, health and meteorological sets are used. The daily number of hospitalizations and deaths due to CVDs are the dependent variables, while daily meteorological variables are the independent variables.

#### 2.1.1. Health Data

Health data are the daily total of CVD hospitalizations and deaths. Daily raw data were provided by the National Institute of Public Health of Québec (Institut national de santé publique du Québec, INSPQ, Québec (QC), Canada). Versions 9 and 10 of the International Classification of Diseases (ICD) were used to classify hospitalization (main and secondary diagnoses), and death (main diagnosis) causes in order to select only the deadliest CVDs for our study (Table 1). The transition from ICD-9 to ICD-10 occurred in April 2006 for the hospitalization files and in January 2000 for the death files. Hospitalizations and deaths were summed to obtain daily numbers of events. Thus, one health datum is either a daily total number of hospitalizations or deaths. Hospitalizations range from 1996 to 2006 (4077 days) inclusively, while deaths range from 1981 to 2011 (11,322 days), resulting in 4077 daily total hospitalizations and 11,322 daily total deaths. Corresponding descriptive statistics are summarized in Table 2.

#### 2.1.2. Meteorological Data

Meteorological data from 1981 to 2011 are provided by Environment Canada and have been matched to deaths and hospitalizations. Daily series of temperature, atmospheric pressure (pressure), relative humidity (humidity), total precipitations, and snow height are available (Table 3). Of note, snow height is measured during the winter, while total precipitations are measured all year round. They were measured over multiple meteorological stations in Montreal and spatially averaged. Giroux et al. [31] found no advantage in using weighted kriging instead of the spatial mean in this area. Corresponding descriptive statistics are given in Table 4. Montreal is in a humid continental climate (http://mddelcc.gouv.qc.ca/climat/normales/climat-qc.htm, accessed on 9 December 2021) where summers are hot and humid while winters can be severely cold.

Maximal values for temperature, humidity and pressure have been used to study the impact of meteorological variables, but mean and minimal values have also been investigated with only minor changes in the results [30]. In this study, maximal values have been chosen to illustrate best how QR can be relevant to public health studies.

### 2.2. Methods

In the rest of this paper, q refers to a quantile order (0<q<1). For a complete description of QR, the reader is referred to the book of Koenker [26]. Let Y be the dependent variable and $X_{1},\ldots X_{p}$ be the independent variables (in this study, the health variables and the meteorological variables, respectively), then the QR model is defined as:(1)$$Q_{y}\left[q|x_{1},x_{2},\ldots ,x_{p}\right]=\beta_{0}\left(q\right)+\beta_{1}\left(q\right)x_{1}+\beta_{2}\left(q\right)x_{2}+\ldots +\beta_{p}\left(q\right)x_{p}$$
for any quantile 0<q<1. In contrast to a classical regression, coefficients $\beta_{0}\left(q\right),\beta_{1}\left(q\right),\ldots ,\beta_{p}\left(q\right)$ vary with q. Their interpretation is similar to that in the classical regression, though only valid in one given quantile. They are obtained by minimizing a sum of weighted absolute residuals [32], whereas classical mean regression is usually solved by ordinary least squares or by maximum likelihood. Unlike the latter, QR does not assume normality or homoscedasticity, which makes QR a more robust regression [33]. Confidence intervals (CI) for estimated coefficients are computed using the inverse rank method [26].

Considered quantile orders in this study spread out from q=0.01 to q=0.99 with a 0.025 step. We used this step in order to explore the health variable conditional distribution in-depth, thus leading to a level of detail that would not be attained by mean-based regression. This study defines a death or hospitalization peak as an observation corresponding to a conditional quantile greater than 90%, as an analogy to the meteorological definition of extreme [34]. Therefore, in the case of our data, death and hospitalizations peaks are defined as days with a total of at least 24 deaths and 172 hospitalizations, respectively (Table 2). The independent variable effects are then compared with the effects in lower conditional quantiles.

Lags are considered for meteorological variables, as their impacts on health can be delayed [16,35,36]. We explored the following lags: 0 (same day exposure), 3, 7, and 14 days. All models controlled for seasonality [37,38], though we did not conduct separate analyses (e.g., summer and winter [39,40]) in order to produce the entire conditional distribution and to not decrease statistical power in the high quantiles. All results are obtained using the statistical software R (R Foundation for Statistical Computing, Vienna, Austria) and *quantreg* package (v5.85). Meteorological variables are standardized (i.e., centered and divided by their standard deviation) in order to facilitate interpretations as the meteorological variables have different measure scales (Table 3). The statistical significance level is set at α=5%.

## 3. Results

QR estimation results are presented in this section for hospitalizations and deaths in Montreal. Results are described for each meteorological variable.

### 3.1. Hospitalizations

Results are shown in Figure 2. QR coefficients in the y-axis are plotted versus quantiles in the x-axis; each column represents a given lag (0, 3, 7, and 14 days) for each meteorological variable. Regarding temperature, the coefficient curve exhibits a similar pattern over the different lags; it starts at a negative value close to 0, decreases until the 10% quantile, increases until the 40% quantile and decreases again up to the highest quantiles (the peaks). Coefficients are non-significant around the 1% and 40% quantiles, whereas in other quantiles an increase in temperature is associated with a decrease in hospitalizations. CI for the classical regression and QR estimations are crossing overall, with the exception of peaks higher than the 95% quantile at lag 7 and 14. In those peaks, differences are significant, and the classical regression estimation underestimates temperature influence. Besides, QR coefficients are the strongest in peaks, more than twice the values for quantiles in the median vicinity.

An inverse association between hospitalizations and humidity is observed at every lag (negative estimated coefficients), except in peaks at lags 3, 7, and 14. However, coefficients are mostly non-significant. There are some exceptions, such as lags 7 and 14, around the median. For peaks (higher than the 95% quantile) at lag 7, the humidity effect is significant with negative estimated coefficients. In this case, a classical regression finds a non-significant effect, whereas there could be a significant one in the peaks. Overall, classical regression coefficients are not significantly different from QR coefficients.

QR coefficients for pressure are neither significant nor different from a classical regression until lag 7. From this lag on, they become significant in quantiles higher than the median. The coefficients are larger in the peaks (particularly at lag 7). Note that coefficients are positive, indicating an increase in hospitalizations when pressure increases at lags 3, 7, and 14. Coefficients are negative the same day, but they are non-significant.

Coefficients for precipitations are mainly non-significant. Some coefficients are, around the 80% quantile. QR and classical regression CI cross each other for every estimation.

The QR coefficient curve for snow is similar to the one for temperature. It follows a decreasing (until the 20% quantile), increasing (until the 40% quantile) and again decreasing pattern. At lags 0 and 3, coefficients are positive and mainly significant, except in peaks higher than the 95% quantile. At lags 7 and 14, the situation is inversed since most of the coefficients become non-significant, whereas those in peaks become so. Note that QR coefficients quantiles around the median are positive but become negative in the peaks. In those peaks, they are furthermore larger in absolute value than in other quantiles, and QR and classical mean regression CI do not cross.

### 3.2. Deaths

Figure 3 provides results for QR estimated coefficients. Regarding temperature, an increase is associated with a decrease in deaths as the estimated coefficients are negative. Moreover, this effect is significant for all quantiles and lags, getting larger (in absolute value) in higher quantiles and lags. QR coefficients in the peaks are indeed 1.5 to 2 times stronger when compared to those in low or median quantiles. Finally, QR coefficients are not different from those estimated by classical regression, except at lag 14. At this lag, classical mean regression coefficients are larger and smaller than those of QR in low and high quantiles respectively.

Humidity coefficients are either constant through the estimated quantiles or decreasing. Humidity coefficients are non-significant on the same day but are significant for the other lags. Estimated coefficients are negative, with the highest intensity in the peaks. Furthermore, the value of the 99% quantile coefficient grows larger from lag 0 to 14 (from −0.5 to −1.5). It is, however, not possible to differentiate QR estimations from those of classical regression.

In contrast to humidity, pressure is significant only for exposure on the same day, between the median and the 95% quantile. Estimated coefficients are negative and not distinguishable from a classical regression. Notice that estimated QR coefficients are the strongest in peaks starting from the 92.5% quantile.

Precipitation coefficients are mainly non-significant. A small portion is significant, under the 20% quantile at lag 14.

Finally, snow estimated coefficients are non-significant at lag 14, though they are for the other lags. Snow is associated with more deaths in peaks, where estimated coefficients are the largest (especially for exposure on the same day). Besides, on the same day and for the peaks higher than the 90% quantile, QR coefficients are different and larger than classical mean regression.

## 4. Discussion

Each meteorological variable is discussed below, followed by general considerations.

### 4.1. Temperature

The temperature turned out to be the most important meteorological variable in terms of statistical significance and coefficient values. A negative association was found as all the estimated coefficients were negative, meaning that lower temperatures are associated with a higher number of CVD hospitalizations and deaths in the investigated quantiles. One possible explanation would be that CVD peaks mainly occur during cold periods [41,42,43]. This negative association between temperature and CVD health has been found in other parts of the world, such as Lille in France [44] or Honk Kong in China [45]. Specifically, in Montreal, it has been shown that found that cold weather induces more CVD deaths than hot weather [46,47]. Furthermore, absolute values for the coefficients increased in higher quantiles, which indicates stronger relationships in the peaks.

### 4.2. Atmospheric Pressure

Opposite results were found concerning the effect of pressure on CVD deaths and hospitalizations. Indeed, an increase in pressure was associated with a decrease in deaths (mainly on the same day, lag 0). Besides, this effect was larger in the peaks than in lower quantiles. Schwartz [48] and Vaduganathan et al. [49] also found a negative association for pressure in the United States and Italy, respectively, though results were for deaths due to all causes for the first and not focusing on the peaks for both. On the contrary, a positive association with deaths due to myocardial infarction and heart failure has been found in previous studies [44,50,51].

For hospitalizations, pressure coefficients were non-significant at lag 0. Starting from lag 3, positive coefficients were then obtained. In Québec, another study found that an increase in pressure in the previous 7 days was associated with a higher risk of being hospitalized for heart failure [51]. Fong and Ma [45] also discovered an association between pressure and CVD hospitalizations in China with lags up to 2 weeks and mentioned that high-pressure systems can produce stagnation episodes (i.e., pollutant accumulation due to weak winds). Notice that atmospheric pressure daily measures were used in this study, but pressure variations also appear to be detrimental to acute myocardial infarctions, especially over one day [52]. Therefore, more studies are needed to explore the relationships between atmospheric pressure and CVD, especially peaks.

### 4.3. Relative Humidity

In this study, relative humidity did not have a significant impact on hospitalizations. This absence of effect has been reported in China and the United States [37,45]. However, Abrignani et al. [53] found a significant relationship between daily hospital admissions due to angina pectoris and relative humidity, although they considered a particular subset of CVDs.

Concerning deaths, a negative and significant association that increased with the lag was found where the effects were stronger for the peaks. Masselot, et al. [46] also found a significant association in Montreal. They uncovered a link between CVD deaths and hospitalizations with humidity at large scales (periodicity of several years). In our study, considered lags spread out from 0 to 14 days, therefore results are not comparable. It is widely accepted that humidity influences perceived temperature, and those two variables are usually studied together in a health context [54,55].

### 4.4. Snow Height

Snow height exhibited inverse relationships for hospitalizations (increasing curve) and deaths (decreasing curve). Furthermore, QR coefficients are significant in the peaks at lags 14 and 0–3, respectively. This suggests an immediate effect of the snow on death peaks but a more prolonged effect on hospitalization peaks. QR coefficients in the peaks are also opposite at the mentioned lags: negative for hospitalizations and positive for deaths. An increase in snow is thus associated with fewer hospitalizations but more deaths when focusing on the peaks. Important and long snowfalls might discourage people from going out, therefore decreasing their exposure [56,57]. On the other hand, there is an increased risk for death and hospitalization due to myocardial infarction immediately after a snowfall in Québec [14], associated with snow shoveling. Those consequences of snowfall are visible in this study around the median, but the negative association in the hospitalization peaks has not been explored yet in the literature.

### 4.5. Precipitations

Precipitation effects were mainly non-significant on CVD deaths and hospitalizations. Some quantiles for hospitalizations (at every lag) revealed significant negative coefficients, which means that an increase in precipitations was associated with a decrease in hospitalizations. However, this effect was non-significant in the peaks. As with the other meteorological variables, this heterogeneous impact would not be revealed using a classical regression. To the best of our knowledge, though some studies link health conditions to precipitations (such as hydric diseases [58]), no study has focused on CVDs. As with humidity, precipitations are rarely investigated alone when studying health impacts, more frequently with temperature or snow height [59,60].

### 4.6. General Considerations

Overall, classical regression and QR coefficients were similar as their 95% CI crossed. Notable exceptions were in the peaks. As QR coefficients varied from one quantile to another, estimated associations were most of the time stronger in the health peaks than in any other quantiles. This coefficient heterogeneity could not be investigated using classical regression and is one of the main advantages of QR. Besides, QR allows reconstructing the entire conditional distribution rather than the conditional mean, as advocated by other authors in health research [61,62,63]. In particular, Siciliani et al. [62] found that the effect of the dependent variables is larger at the higher conditional quantiles, as found in the present study (e.g., temperature effect in Figure 2). 

Another difference that can be observed when using QR is the sign inversion for QR parameters. For instance, Marrie et al. [61] found opposing effects between a mean regression and a 90% QR. This situation occurred in this study, such as the case of snow with hospitalizations at lag 14 (Figure 2). QR coefficients around the mean were estimated positive, whereas they were negative starting from the 90% quantile, thus opposing the classical mean regression. Therefore, QR is relevant in the health peak study as it allows comparing the conditional mean to the high quantiles, where results may be larger or even inverse. More generally, it complements classical regression analysis by providing a conditional distribution [64].

In fields other than environmental health, heterogeneous and accentuated effects for independent variables in the high conditional quantiles have also been observed (climatology or hydrogeology [65,66]). Authors support QR use as it can decompose relationships in quantiles other than the median. More recently, QR has been used in the same spirit to study the effects of home quarantine during the COVID-19 pandemic [67]. The authors evaluated multiple variables on the distribution of happiness and found that low and high quantiles were not influenced by the same variables. In other words, people with increased happiness (high quantiles) did not experience the quarantine at home the same way that people with decreased happiness (low quantiles) did.

### 4.7. Limitations

Daily aggregation for the health variables did not allow for the inclusion of relevant individual variables, such as comorbidities or life habits (smoking, nutrition, or physical exercise). Those variables have an impact on CVD occurrence [7,8,9]. Furthermore, besides meteorological variables, atmospheric pollution variables should also be included in future research [68,69].

From a statistical point of view, nonlinear QR could be used to investigate specific quantiles [61]. However, since nonlinear QR produces a regression curve instead of a regression line for each quantile, chances that multiple curves cross each other would be increased (a non-trivial issue known as crossing QR, for which research is still active [70,71]), leading to invalid results.

## 5. Conclusions

This paper aimed at studying relationships between CVD health and meteorological conditions using QR, examining health peaks and lower conditional quantiles at the same time. Different lags and multiple quantiles have been investigated. Even though the focus here is on CVDs, QR could also be used to investigate possible relationships between meteorological conditions and any other chronic disease.

In previous studies, the temperature has shown a significant effect. This is similar to the results in this study, confirming the importance of temperature on CVD health. Moreover, QR has allowed for more precise estimations and has shown the heterogeneous effects of meteorological conditions in the health variables quantiles. In particular, impacts were found to be stronger in health peaks compared to median quantiles. This result could not be obtained using a classical regression, as the information would be “diluted”. Therefore, along with meteorological forecasting, those results could be useful for health peak forecasting and its management in health warnings and services.

Finally, this study supports the idea of completing classical health and meteorological conditions studies with health peaks studies. Doing so would result in more complete estimations of health variable conditional distributions.

## Figures and Tables

**Figure 1 ijerph-18-13277-f001:**
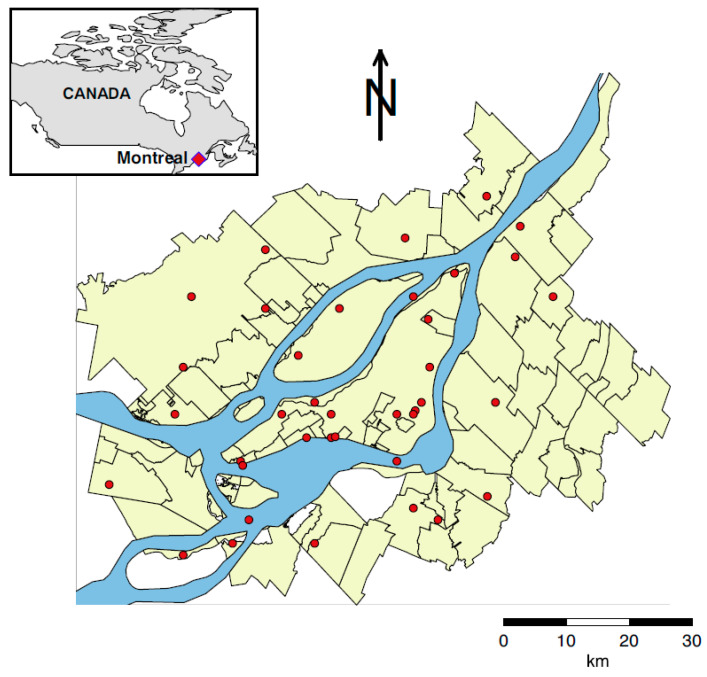
Montreal metropolitan community (Canada). Red dots are the meteorological stations used for measuring meteorological variables.

**Figure 2 ijerph-18-13277-f002:**
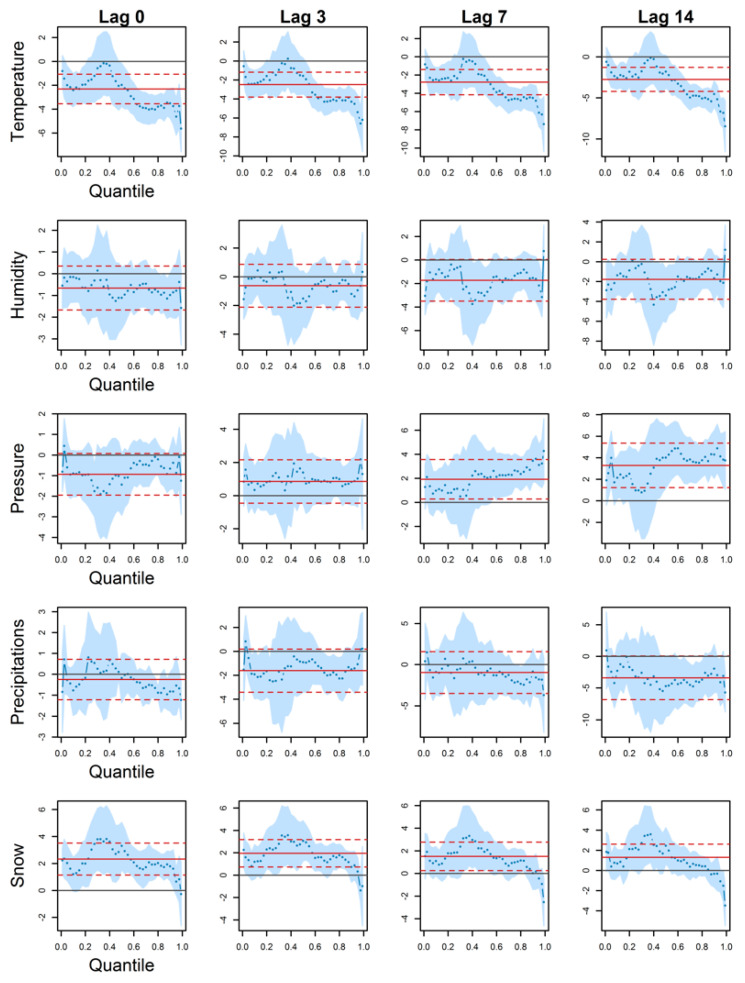
QR (blue dots) and classical mean regression (red full line) coefficients against quantiles, for hospitalizations in Montreal. 95% confidence intervals are shown in light blue for QR and in red dashed lines for classical mean regression.

**Figure 3 ijerph-18-13277-f003:**
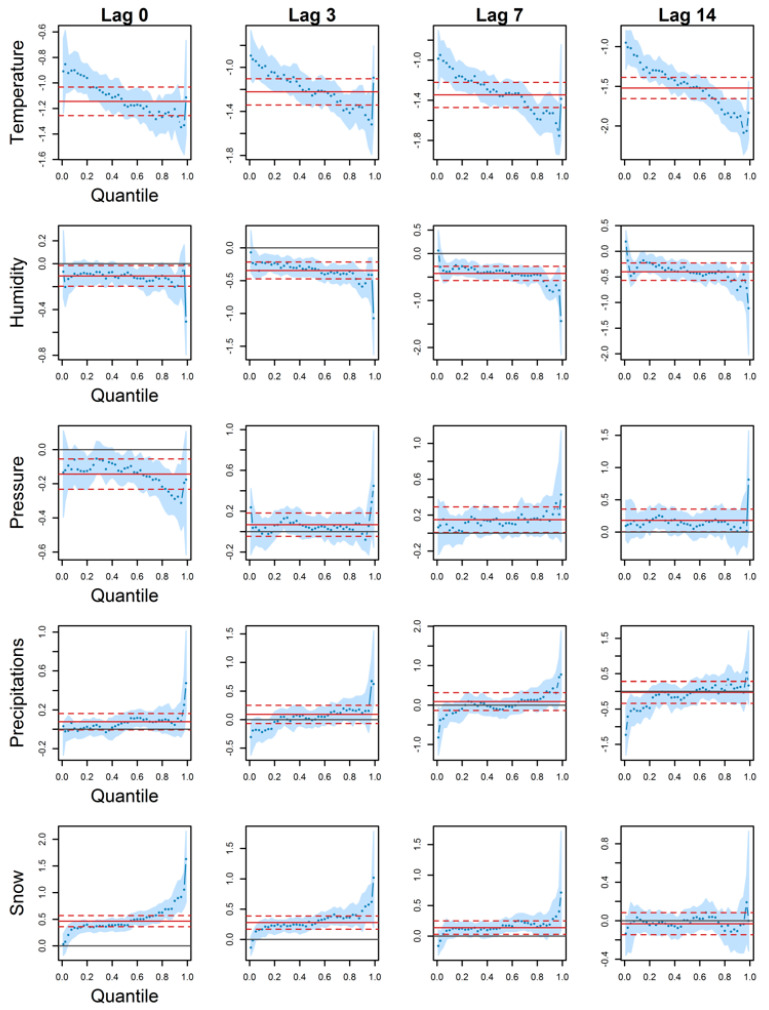
QR (blue dots) and classical mean regression (red full line) coefficients against quantiles for deaths in Montreal. 95% confidence intervals are shown in light blue for QR and in red dashed lines for classical mean regression.

**Table 1 ijerph-18-13277-t001:** ICD-9 and ICD-10 for considered CVDs in this study.

Most Deadly CVDs	ICD-9	ICD-10
Ischemic heart diseases	410–414	I20–I25
Heart failure	428	I50
Cerebrovascular diseases Transient cerebral ischemia	362.3	G45.x (excluding G45.4)
	430	H34.0
	431	H34.1
	434.x	I60.x
	435.x	I61.x
	436	I63.x (excluding I63.6)
		I64

**Table 2 ijerph-18-13277-t002:** Descriptive statistics for daily CVD deaths and hospitalizations in Montreal.

	Deaths	Hospitalizations
Minimum	3	49
Maximum	53	220
Mean	17	131
Median	17	136
75% quantile	20	158
90% quantile	24	172

**Table 3 ijerph-18-13277-t003:** Meteorological variables description.

Variable	Type	Unit
Maximal temperature	Daily data	Celsius degrees
Total precipitations		Millimeter
Snow height		Centimeters
Maximal atmospheric pressure	Hourly data	Kilopascals
Maximal relative humidity		Percentages (%)

**Table 4 ijerph-18-13277-t004:** Descriptive statistics for meteorological variables in Montreal from 1981 to 2011.

	Temperature	Humidity	Pressure	Precipitations	Snow
Minimum	−26.5	34.0	99.1	0.0	0.0
Maximum	35.0	100.0	105.2	89.8	79.2
Mean	11.4	81.6	102.0	2.8	6.8
Median	12.2	84.0	101.9	0.3	0.0
75% quantile	22.4	91.8	102.5	3.0	8.9
90% quantile	26.8	96.0	103.0	8.9	27.4

## Data Availability

Data are not available, due to governmental privacy policy.

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
