# Peer review of "Cardiovascular Health Peaks and Meteorological Conditions: A Quantile Regression Approach"

_ijerph, 2021, doi:10.3390/ijerph182413277_

Round 1

Reviewer 1 Report

Authors propose the quantile regression approach for the assessment of health peaks. The approach is interesting since it is based on quantiles rather than mean values and it seems well suited in case of asimmetric variable distribution. The main focus of the paper seems to be the identification of health peaks relevant from a public health point of view in the context of the association between cardiovascular outcomes and meteorological variables. The health peaks are in my opinion a matter for different aims and it is unclear which one the authors are interested to. The first deals with surveillance activities, requiring a real-time data source which can be investigated to assess timely health peak of public health relevance. In this case, no exposure is needed in the identification of the health peak. In the case of an epidemiological time-series such as that analysed here (meteo variables and health outcomes), another aim could be more to compare the health outcomes in extreme meteorological event vs non extreme meteorological conditions (i.e. heat wave vs non heat wave). Alternatively, another aim could be to take into account outliers (i.e. extreme health peaks) in the time-series analysis of an exposure (in this case meteo variables) and of an outcome (i.e. CVD ones). If this is the case, quantile regression can be viewed as a method to better take into account of data variability, something similar to smoothing techniques, but at the end it should lead to association estimates which are not provided in the paper that seems only an example of methods application rather than a study interested to investigate a specific association.

After clarifying the objective some of these issues will probably lapse.

  1. If the interest is to assess this specific epidemiological association, the control only for seasonality seems weak and it would be important to stratify by warm and cold season.
  2. The linear regression (i.e. Gaussian) as comparison method similarly seems not the best for count data such as mortality and hospital admission counts which usually are investigated with Poisson distribution.
  3. Table 2 needs to be improved by stratifiyng by season or specific months and by adding information about health peaks.

Reviewer 2 Report

Suggestions to add comments and discussion related to health data as meteorological variables 

Add more precise definition/explanation of health peaks

Reviewer 3 Report

Finally! I write down 'finally' because this team, proposing the manuscript, approached the problem with the right and deep statistical sense: it is surely not the 'average' which killed you in short term periods, and the analysis performed in the manuscript clearly demonstrated the importance of a proper appliance of statistical methodologies. I absolutely agree with the authors that classical regression 'diluted' the information and results are misleading.

The paper is very well presented (with the exception of the one point reported in my specific note). Data, methodology and results are very well presented and very useful to a wide audience. Discussion and conclusions are strongly effective considering the findings. Because of the note content, I must place "minor revision"

  • note: never write down again "hotter temperature" like on page 10 line 245. Temperatures are high or low, never hot or cool.

Reviewer 4 Report

What do you mean by health peaks? The focus of the study is cardiovascular disease so why not the term 'cardiovascular disease' included in the research title.

The literature review should specifically mention some examples of increased cardiovascular disease with meteorological data.

The rationale for doing the research is also lacks clarity.

Is table 2 represent deaths only due to cardiovascular disease? How are you so sure about it?

It is not clear whether you used the meteorological data for the last 10 years or not.

Table 4 presents descriptive statistics for meteorological data from Montreal. How many years of data was analyzed?

Mention the limitations of the research in the discussion part.

Round 2

Reviewer 4 Report

thanks for making the changes.